# Vitamin D–VDR Novel Anti-Inflammatory Molecules—New Insights into Their Effects on Liver Diseases

**DOI:** 10.3390/ijms23158465

**Published:** 2022-07-30

**Authors:** Ioanna Aggeletopoulou, Konstantinos Thomopoulos, Athanasia Mouzaki, Christos Triantos

**Affiliations:** 1Division of Gastroenterology, Department of Internal Medicine, University Hospital of Patras, GR-26504 Patras, Greece; iaggel@hotmail.com (I.A.); chtriantos@hotmail.com (C.T.); 2Division of Hematology, Department of Internal Medicine, Medical School, University of Patras, GR-26504 Patras, Greece; mouzaki@upatras.gr

**Keywords:** vitamin D, VDR, signaling, liver disease, HBV, HCV, AIH, NAFLD, PBC

## Abstract

There is consistent evidence that vitamin D deficiency is strongly associated with liver dysfunction, disease severity, and poor prognosis in patients with liver disease. Vitamin D and its receptor (VDR) contribute to the regulation of innate and adaptive immune responses. The presence of genetic variants of vitamin D- and VDR-associated genes has been associated with liver disease progression. In our recent work, we summarized the progress in understanding the molecular mechanisms involved in vitamin D–VDR signaling and discussed the functional significance of VDR signaling in specific cell populations in liver disease. The current review focuses on the complex interaction between immune and liver cells in the maintenance of liver homeostasis and the development of liver injury, the interplay of vitamin D and VDR in the development and outcome of liver disease, the role of vitamin D- and VDR-associated genetic variants in modulating the occurrence and severity of liver disease, and the therapeutic value of vitamin D supplementation in various liver diseases. The association of the vitamin D–VDR complex with liver dysfunction shows great potential for clinical application and supports its use as a prognostic index and diagnostic tool.

## 1. Introduction

The liver is the largest solid organ in the human body and it performs a variety of functions, including systemic metabolism, glucose and lipid synthesis, immunity, and detoxification [1]. Liver homeostasis is maintained by several mechanisms, particularly the suppression of immune responses and the development or maintenance of immune tolerance [1]. The spleen is the largest secondary lymphoid organ in the human body, and beyond its function in hematopoiesis and the clearance of red blood cells it is involved in a variety of immunological functions, such as the regulation of inflammation-related immune responses [2,3]. The enlargement of the spleen is a common feature in patients with nonalcoholic fatty liver disease (NAFLD) or hepatic steatosis, which represents the so-called liver–spleen axis [4,5]; the liver–spleen axis is closely related to inflammatory outcomes resulting from the interplay between nutrition and inflammation [5].

Hepatic macrophages are responsible for maintaining liver homeostasis and play a critical role in the development and progression of liver disease. These cells are mainly derived from Kupffer cells (KCs), infiltrating macrophage populations, or monocyte-derived macrophages (MoMFs) stored as reservoirs in the spleen [6]. Previously, macrophages were classified into a pro-inflammatory M1 phenotype and an alternative, anti-inflammatory M2 phenotype [7]. However, in recent years, considerable heterogeneity has been observed between hepatic macrophage populations with different functions and gene signatures [6]. Although KCs represent the major hepatic macrophage population involved in homeostasis under normal conditions, massive infiltration of MoMFs into the damaged liver is observed after metabolic or toxic injury to the liver [6].

In this review, we focus on the complex interaction between immune and liver cells in the maintenance of liver homeostasis and the development of liver injury, discuss advances in the molecular mechanisms of vitamin D–VDR signaling in the context of liver disease development and progression, present the recent findings on the role of vitamin D and VDR-associated genetic variants in regulating the occurrence and severity of liver disease, and finally summarize the therapeutic value of vitamin D supplementation.

## 2. Materials and Methods

### 2.1. Search Strategy

A comprehensive literature search of the PubMed, MEDLINE, and COCHRANE library databases for articles from inception to 31 May 2022 was conducted to identify eligible studies. The Boolean operators AND and OR and NOT were used to combine the search terms and focus the search results because the topic under study contained multiple search terms. The key search terms consisted of the words “vitamin D”, “25(OH)D”, “1,25(OH)2D3”, “vitamin D receptor”, “VDR”, “hepatitis B”, “HBV”, “hepatitis C”, “HCV”, “autoimmune hepatitis”, “AIH”, “nonalcoholic fatty liver disease”, “NAFLD”, “primary biliary cholangitis”, and “PBC” in various combinations. Two authors (I.A. and C.T.) independently screened and reviewed the titles and abstracts of the eligible studies. Discrepancies were resolved via consultation within the research group. Additional relevant articles that were cited in other documents but did not appear in the original search were identified and included.

### 2.2. Selection of Studies

Studies published in peer-reviewed English-language journals and available as full texts online or through a library were considered. Randomized controlled trials; cohort, multicenter, comparative, and observational studies; clinical trials; review articles; systematic reviews; meta-analyses; case series; and case reports were included.

## 3. Hepatocytes and Liver Damage

Signals from the microenvironment determine the migration and polarization of macrophages toward pro-inflammatory or anti-inflammatory responses. The interplay between KCs, liver sinusoidal endothelial cells (LSECs), and dendritic cells (DCs) is of great importance for the initiation and regulation of liver immune responses through antigen presentation and the production of various cytokines and chemokines in association with T and B lymphocytes, natural killer (NK) cells, and neutrophils [8]. The activation of an inflammatory response by hepatic stellate cells (HSCs) and KCs leads to the infiltration of neutrophils, monocytes, and NK and natural killer T (NKT) cells. The recruitment of MoMFs and the interplay between HSCs and macrophages are critical for the progression or pathogenesis of liver disease and tissue regeneration after liver injury [6]. The balance between pro-inflammatory and anti-inflammatory T cell populations and the distinct functions of macrophages and DCs are responsible for the outcome of the immune response in the liver [8]. An imbalance in this mechanism can lead to liver inflammation and fibrosis.

Liver fibrosis is a reversible wound healing response to chronic or acute cellular injury [9]. Hepatocytes, HSCs, liver macrophages, cholangiocytes, LSECs, and immune cells have been implicated in the pathogenesis of liver fibrosis [10]. In particular, the activation of HSCs and the HSC-mediated deposition of extracellular matrix (ECM) proteins, especially type I collagen, leads to the destruction of the liver architecture and liver cell dysfunction [9]. In the healthy liver, HSCs are in a dormant state and form the main store of vitamin A in the body. After liver injury, HSCs differentiate into activated proliferative myofibroblasts with an increased accumulation of ECM, leading to the secretion of pro-inflammatory cytokines and pro-fibrotic growth factors [11]. In parallel, hepatic myofibroblasts induce hepatic macrophage differentiation, which further enhances the pro-inflammatory and pro-fibrotic responses [11]. KCs are involved in the initial response to injury by secreting cytokines and chemokines and recruiting monocytes by producing the chemokine C-C motif chemokine ligand 2 (CCL2) and CCL5 [12,13]. Fibrosis can be further enhanced by MoMFs that release factors such as transforming growth factor-β (TGF-β), interleukin 1β (IL-1β), platelet-derived growth factor (PDGF), and CCL2, activating hematopoietic stem cells (HSCs) and inducing inflammation [6]. PDGF has been identified as the most potent mitogenic factor for HSCs, whereas TGF-β is a stimulatory signaling pathway responsible for the deposition of ECM proteins, and may influence the regulatory process mentioned above [14,15].

## 4. Physiology and Metabolism of Vitamin D

Vitamin D is a fat-soluble vitamin that occurs in two different chemical structures: vitamin D_2_ or ergocalciferol and vitamin D_3_ or cholecalciferol. Both vitamin D_3_ and vitamin D_2_ are synthesized in the skin via the photochemical conversion of 7-dehydrocholesterol to pre-vitamin D or are absorbed in the small intestine [16,17]. These forms are considered biologically inactive until being enzymatically hydroxylated [18]. Vitamin D_3_ is transferred to the liver after binding to vitamin D binding protein (DBP). In the liver, the vitamin D-DBP complex undergoes 25-hydroxylation, resulting in the formation of 25-hydroxyvitamin D or 25(OH)D [18,19]. Furthermore, 25(OH)D is the most abundant circulating vitamin D metabolite and is used to determine the vitamin D status. Subsequently, this form is further hydroxylated in the kidneys, leading to the formation of the biologically active form of vitamin D, 1,25(OH)2D3 [18,19]. This process is regulated by calcium and phosphate, parathyroid hormone [20], and fibroblast growth factor [16]. The formation of 1,25(OH)2D3 by 1-α-hydroxylase (CYP27B1) occurs not only in renal tissues but also in extra-renal tissues, including cells of the immune system and intestinal macrophages [21]. The activation of various cytokines or Toll-like receptors (TLR) in macrophages and T-cell receptors (TCR) in T cells is essential for CYP27B1 induction in immune cells [21,22,23,24].

Beyond the classical functions of vitamin D in bone mineralization and systemic calcium homeostasis, it has recently been reported to exert pleiotropic biological functions in various cells and tissues, regulating immune modulation, cell proliferation, and differentiation [18,19]. In parallel, the role of vitamin D in the hepatic pathophysiology has been investigated, showing that it can inhibit fibrosis in liver tissue [25,26]. Several studies have associated vitamin D deficiency with liver dysfunction, demonstrating an inverse relationship between 25(OH)D levels and the severity of liver disease [17,26,27,28,29,30,31,32,33,34,35,36].

## 5. Vitamin D Receptor (VDR)

The 1,25(OH)2D3 binds and induces the transcription factor VDR [37,38], which directly affects gene regulation [38]. VDR belongs to the nuclear receptor superfamily [39] and acts on many genes in about half of the cell types and human tissues [40]. Therefore, the biological functions of vitamin D_3_ in health and disease are directly related to 1,25(OH)2D3-dependent changes in the transcriptome in VDR-expressing cells [41]. Upon vitamin D activation, a conformational rearrangement of the VDR occurs, allowing its heterodimerization with the retinoid X receptor (RXR). Subsequently, this complex migrates to the nucleus, and after binding to specific genomic sequences (vitamin D response elements, VDREs) in the promoter, modulates the transcription of target genes. The ligand-bound VDR provides a mechanistic basis for the activation or repression of specific target genes by vitamin D signaling or by heterogeneous mechanisms, many of which involve interactions with various transcription factors [42].

Numerous nuclear receptors, including VDR, and their interplay have been linked to the development of liver disease, suggesting that vitamin D may influence disease by targeting VDR or other receptors [43]. At the same time, there is increasing evidence that a subset of plasma-membrane-bound VDRs elicit rapid biological effects through nongenomic pathways [44].

VDR is expressed in vitamin D target tissues and immune cells [45,46]. In particular, VDR is expressed in activated T cells and DCs. In addition, VDR is also expressed in B cells, cytotoxic NK cells, and other antigen-presenting cells (APCs), such as macrophages and monocytes [45,46,47,48]. The relationship between VDR and inflammatory responses has been demonstrated in several types of immune cells [49,50]. The 1,25(OH)2D3-VDR complex has been shown to exert an inhibitory effect on DC maturation, resulting in a different phenotypic state that is resistant to maturation stimuli [49]. This process appears to be associated with various antigen-specific immune responses in vivo and may be involved in the regulation of immune homeostasis [49]. In parallel, vitamin D–VDR has been implicated in TCR signaling and the stimulation of T cells via the p38 mitogen-activated protein kinase (MAPK) signaling pathway [50]. This pathway has been shown to upregulate the production of T regulatory cells (Tregs) and the gene expression of specific cytokines (IL-4, IL-10, and TGF-β), while suppressing the differentiation of CD4+ T helper cells into T helper 1 (Th1) cells, which further inhibits the effector functions of the latter [51].

In the healthy liver, the VDR expression is low [52]. However, strong expression of VDR has been detected in nonparenchymal liver cells, including KCs, HSCs, and biliary epithelial cells [46]. The nonparenchymal cells, such as HSCs, play an essential role in liver fibrosis because they are the main source of ECM components [53]. Therefore, the liver response to vitamin D is thought to occur via the nonparenchymal cells [44].

## 6. Vitamin D, VDR, and Liver Diseases

Vitamin D deficiency and genetic polymorphisms are commonly associated with the onset, natural history, and outcome of several diseases, including chronic hepatitis B or C viral infections and autoimmune diseases. The role of vitamin D deficiency in chronic liver disease is not fully understood. The recent evidence suggests that vitamin D is involved in signaling pathways that control the expression of antiproliferative, anti-inflammatory, and antifibrotic genes, and consequently plays an essential role in the natural history of chronic liver disease [17,42,54]. In parallel, the vitamin D/VDR axis appears to be a critical modulator of the development and severity of a variety of liver diseases [17,42,44,54].

## 7. Possible Mechanisms of the Association between Vitamin D–VDR and Liver Disease

### 7.1. Vitamin D–VDR and Hepatitis B Virus (HBV)

Vitamin D deficiency has been demonstrated in patients with untreated, active chronic hepatitis B (CHB) [55], a condition associated with unfavorable clinical outcomes, including hepatocellular carcinoma (HCC) and increased liver-related mortality [56]. The data support that CHB increases the risk of vitamin D deficiency. One study demonstrated that long-term treatment with nucleoside or nucleotide analogs significantly increased the 25(OH)D levels in patients with undetected HBV-DNA [57]. In addition, low serum vitamin D levels have been associated with increased HBV replication [55,56,57,58,59], suggesting that vitamin D suppresses this process. However, a recent study suggested that vitamin D has no effect on HBV replication rates because the VDR levels were decreased in the presence of the virus, attenuating vitamin D signal transduction [60]. It is possible that HBV suppresses the immune activation by downregulating the vitamin D signal transduction, as it has been shown that VDR expression was suppressed in an HBV-infected hepatic cell line, preventing the effect of vitamin D on the viral transcription and production [5]. Alternatively, HBV may use the above mechanism to evade the immune system by affecting the expression of immune modulators such as antimicrobial peptide (CAMP), cathelicidin, and tumor necrosis factor α (TNF-α) (Figure 1) [60]. 

In humans, triggering TLR1, 2, 4 leads to the production of the active form of vitamin D, the binding of vitamin D to VDR/RXR, and the recruitment of this complex to VDREs. This process leads to the transcription of mRNAs encoding various proteins such as cytokines, cathelicidin, and defensins [23,61]. Cathelicidin is a positively charged protein that degrades the viral envelope, as does bacterial degradation in macrophages [62,63]. Defensin serves as a chemoattractant for T cells [62,63]. Th1 cells are suppressed by Tregs and VDR activation leads to the secretion of IL-2 and TNF-α by macrophages [64]. Tregs increase the expression of IL-10 and TGF-β, which in turn inhibit Th1 cells [65,66]. Th1 cells downregulate the expression of IL-2, inhibiting cellular responses. The expression of these cytokines also inhibits antibody production by B cells and induces Th2 cells to increase the synthesis of IL-4, IL-5, IL-10, and IL-13. This process results in a systemic shift of the immune system toward a Th2 response [64]. The antibodies produced by plasma cells bind to the viral antigens and infected hepatocytes, leading to viral destruction [67] (Figure 1).

Several studies have also investigated the role of genetic VDR variants in the outcome of HBV infection [68]. A recent meta-analysis showed no association between VDR TaqI, ApaI, and BsmI polymorphisms and HBV infection [68], whereas FokI FF, Ff genotype, and F alleles increase the risk of HBV infection [68]. It is possible that the F (C) allele tends to have higher transcriptional activation, leading to an increase in VDR expression, which in turn could bind more 1,25(OH)2D3, leading to a decrease in IL-12 and IFN-γ expression and an induction of IL-4 and IL-10 expression [68]. These molecular changes may inhibit the activity of T helper effector cells, resulting in impaired HBV clearance [29].

#### Vitamin D Supplementation and HBV

Adequate vitamin D levels are closely related to the development of effective vaccine responses against HBV. The influence of vitamin D supplementation on the course of short- and long-term HBV vaccination was investigated in a recent study [69]. This showed that vitamin D levels were positively associated with anti-HBsAg IgG levels and the percent avidity in patients receiving vitamin D supplementation [69]. In contrast, another study examining the effect of oral vitamin D_3_ supplementation on HBV vaccination showed that vitamin D supplementation starting 3 days after vaccination had no effect on the vaccination response [70].

### 7.2. Vitamin D–VDR and Hepatitis C Virus (HCV)

Recent studies have shown that vitamin D plays an important role in the persistence and clearance of HCV infection by suppressing HCV replication through several mechanisms, including the induction of oxidative stress pathways, the utilization of IFN-based signaling pathways, the enhancement of host chemotactic factors and autophagic machinery, and the enhancement of zinc uptake [44].

From a mechanistic perspective (Figure 2), vitamin D binding to VDR modulates the inflammatory response triggered by HCV infection by inhibiting the gene expression of pro-inflammatory cytokines and the TLR/nuclear factor-κB (NF-κB) signaling pathway [71]. HCV pathogenicity is modulated by calcitriol via the stimulation of VDR and blockade of PPAR-α/β/γ activity [72,73]. The suppressive effect of calcitriol on PPAR-α/γ leads to the suppression of viral infections; the suppressive effect on PPAR-β/γ leads to a decrease in nitrative stress, and finally the suppressive effect on PPAR-γ leads to a decrease in lipid accumulation [72]. In parallel, the suppression of apolipoprotein J (apoJ) and glucose-regulated protein 78 (Grp78) by the endoplasmic-reticulum-associated protein degradation (ERAD) caused by calcitriol leads to the inhibition of HCV [72] (Figure 2).

Considering that vitamin D signaling is closely associated with infectious liver diseases, the use of vitamin D metabolites as therapeutic agents has been investigated. The recent results have shown that treatment with 1,25(OH)2D3 suppresses HCV in Huh 7.5 cells infected with HJ3-5 and JFH-1 HCV [73,74,75]. Another study also found a decrease in HCV core antigen production in an in vitro system under 25(OH)D3 treatment at both extracellular and intracellular levels [76]. The inhibition of HCV replication by vitamin D derivatives, including vitamin D2, vitamin D3, and 1,25(OH)2D3, was observed in vitro [72,77]. Saleh et al. [78] investigated the mechanism of selected calcitriol analogs in regulating the hepatocellular transcriptome and suppressing HCV, and reported that the structurally related vitamin D analogs calcipotriol and tacalcitiol, but not calcitriol itself, inhibited HCV replication via a VDR-dependent pathway. Specifically, critical regulatory elements of the VDR bind to calcipotriol through distinct local interaction patterns that affect key functional residues and elicit stronger regulatory effects on the hepatocyte and macrophage transcriptome [78]. Conversely, another study showed that the effect of vitamin D3 is not mediated by its metabolic conversion to calcitriol, but may be due to its primary metabolite 25(OH)D3 [75]; it was shown that the antiviral activity of vitamin D_3_ and 25(OH)D3 was not impaired in a VDR knockout animal model, suggesting that the anti-hepatitis C effect of 25(OH)D3 is VDR-independent [75] (Figure 2). The data suggest a role for vitamin D in the response to IFN-α-based HCV treatment. Specifically, calcitriol enhanced the IFN-α-induced binding of phosphorylated signal transducer and activator of transcription 1 (STAT1) to its DNA target sequences, and the knockdown of VDR resulted in an enhanced hepatocellular response to IFN-α treatment, highlighting VDR as a novel suppressor of IFN-α-induced signal transduction via the Janus kinase (JAK)–STAT pathway [79] (Figure 2). This enhancement of antiviral IFN-α activity by 1,25(OH)2D3 appears to be mediated by the nongenomic activity of vitamin D [79]. This may explain the association between vitamin D deficiency and a poor response to PEG-IFN-α treatment [80].

A meta-analysis by Kitson et al. [81] evaluated 11 studies and showed no statistically significant difference in circulating vitamin D levels between HCV-infected patients who achieved SVR in response to PEG/IFN/ribavirin therapy and those who did not. A recent meta-analysis evaluating 28 studies for possible associations between vitamin D levels and the severity of liver inflammation in HCV-infected patients yielded conflicting results [82]. Specifically, the results showed that low vitamin D levels were significantly associated with an increased likelihood of HCV infection and that in HCV-infected patients, low vitamin D levels were associated with severe fibrosis, advanced inflammation, and nonresponse to antiviral treatment. In addition, vitamin D contributes to the regulation of inflammation by suppressing the nuclear translocation of NF-κB, thereby reducing the production of inflammatory cytokines [13].

Finally, the HALT-C study [78] examined the effects of 40 genetic variants in genes controlling vitamin D metabolism or the VDR/TGF-β1/SMAD3 interaction on chronic HCV disease progression. The results showed that 11 polymorphisms were potentially associated with liver-related outcomes, whereas only one SNP in the TGF-β1 gene was associated with hepatic decompensation in HCV patients [83].

#### Vitamin D Supplementation and HCV

Two vitamin D metabolites, 25-hydroxyvitamin D3 [25-(OH)D3] and 1α,25-dihydroxyvitamin D3 [1α,25-(OH)2D3], have been reported to have anti-HCV effects [84]. Short-term vitamin D supplementation after treatment with direct-acting antivirals (DAAs) for chronic hepatitis C did not show an improvement in serum fibrogenic markers and may not promote an improvement in residual liver fibrosis [85]. The effect of vitamin D supplementation has been studied as add-on therapy to IFN-α/ribavirin (RBV) in the treatment of patients with chronic HCV, suggesting that vitamin D supplementation improves the sustained virologic response (SVR). The SVR in patients receiving natural vitamin D3 (cholecalciferol) was dependent on HCV genotypes, ranging from 54% to 86% in HCV genotype 1 [86,87] and up to 95% in genotypes 2 and 3 [88]. Vitamin D supplementation in combination with IFN-α/RBV therapy in patients undergoing liver transplantation and receiving immunosuppressive treatment significantly improved the SVR rates [84]. Vitamin D in combination with Peg-IFN-α/RBV significantly improved the early viral response rates in treatment-naive HCV genotype 1, 2, 3, or 4 patients [89]. In a meta-analysis comparing patients with or without vitamin D supplementation regardless of HCV genotype, a high SVR odds ratio (OR = 4.6, 95% CI: 1.7–12.6) was documented [90]. These results were confirmed by a meta-analysis of patients with chronic hepatitis C and different HCV genotypes [91]. 

Contradictory results were shown in a study investigating the effect of vitamin D supplementation to PEG/RBV therapy in treatment-naïve HCV patients with a genotype of 1/4, as no effect was reported in terms of an early, rapid, and sustained virologic response [92]. In parallel, Ladero et al. [93] showed no significant effect of vitamin D supplementation on viral or biochemical HCV variables. Esmat et al. [94] reported no beneficial effect of vitamin D supplementation on SVR in HCV genotype 4 patients. In a multicenter randomized controlled trial, patients with chronic hepatitis C were randomly divided into two groups: PEG-IFN-α plus RBV and PEG-IFN-α plus RBV + vitamin D. The results showed that the vitamin D addition had no effect on SVR rates in treatment-naïve patients, regardless of the genotype [95]. A recent review showed that the effect of vitamin D on the early, rapid, and sustained virologic response in chronic hepatitis C patients is highly uncertain compared with placebo or no treatment [96].

In addition to the use of natural vitamin D supplements, other forms of vitamin D have been used in other studies. Kondo et al. [97] studied 1α-hydroxyvitamin D3 (1α-(OH)D3) in conjunction with Peg-IFN-α/RBV treatment and demonstrated an improvement in treatment efficacy. The effect of the 1α-(OH)D3 supplementation compared with vitamin D3 was investigated, and it was found that the vitamin-D3-supplemented group showed a more rapid decline in serum HCV levels than the 1α-(OH)D3 group in the initial treatment phases; however, the final SVR rate did not show a significant difference between the groups [98].

A recent study demonstrated an association between IFN supplementation and changes in 25(OH)D levels in response to IFN-based treatment in HCV-positive patients [99]. The activation of VDR signaling in PBMCs was associated with increased 25(OH)D levels, highlighting the importance of IFN-α, circulating vitamin D levels, and VDR signaling in modulating innate immunity, with a focus on PBMCs [99].

### 7.3. Vitamin D–VDR and Autoimmune Hepatitis

There is increasing evidence of a link between low vitamin D levels and autoimmunity. In autoimmune hepatitis (AIH), a deficiency of 25(OH)D3 (less than 10 ng/mL) has been associated with severe inflammatory activity of the liver or interface hepatitis, progressive liver fibrosis, and treatment failure [100].

A recent study demonstrated an association between low vitamin D levels and increased inflammatory and stress responses, decreased T-cell subpopulations, and impaired immunity in AIH patients [101] (Figure 3). In contrast, another study reported that VDR signaling leads to a rapid increase in reactive oxygen species (ROS), which may adversely affect the liver in AIH [102] (Figure 3).

#### Vitamin D Supplementation and AIH

Vitamin D supplementation in patients with AIH has been advocated [100] because it has been demonstrated that circulating 25-hydroxyvitamin D levels are low in the majority of patients with AIH and that they have an increased nonresponse rate to steroids compared with healthy individuals [103]. Vitamin D supplementation has been shown to reduce inflammation and limit liver injury in patients with AIH, acting mainly as an antioxidant [104]. Vitamin D also has great potential to improve the glucocorticoid response or induce a dose reduction. In addition, vitamin D analogues with low calcemic effect, high efficacy, and independence from liver hydroxylation have great potential as alternative treatment strategies for the management of AIH [104].

### 7.4. Vitamin D–VDR and Primary Biliary Cholangitis (PBC)

Genetic studies have shown that there are potential proteins linking vitamin D and the PBC pathology, such as the major histocompatibility complex class II (MHC II), the VDR, TLRs, apolipoprotein E, Nramp1, and cytotoxic T lymphocyte antigen-4 [105]. The effect of vitamin D on PBC could also potentially be through molecular mechanisms such as matrix metalloproteinases, prostaglandins, reactive oxygen species, and TGF-β cell signaling [105].

One proposed mechanism for attenuating inflammatory responses through vitamin D–VDR signaling is through influencing the miRNA155-SOCS1 (suppressor of cytokine signaling 1) axis [106]. The expression of VDR/miRNA155-modulated SOCS1 was decreased in PBC patients, and it has been suggested that this leads to inadequate negative regulation of cytokine signaling [107] (Figure 4).

#### Vitamin D Supplementation and PBC

Recent guidelines from the European Association for the Study of the Liver (EASL) recommend the use of calcium and vitamin D supplements in patients with PBC, depending on local practices [108]. A study investigating the effect of calcitriol supplements in PBC patients found that patients receiving supplementation had significantly higher 25(OH)D3 levels and a better prognosis than patients not receiving calcitriol [109].

### 7.5. Vitamin D–VDR and NAFLD

Several studies have examined the effects of vitamin D supplementation on insulin resistance and glucose metabolism in patients with NAFLD, but although the data from in vivo and in vitro studies support an association between vitamin D deficiency and NAFLD, causality has not yet been established.

The role of vitamin D supplementation in controlling blood glucose levels, insulin sensitivity, and the progression of NAFLD is not fully understood. One proposed mechanism is that vitamin D treatment leads to increased transcriptional regulation and the translocation of glucose transporter 4 (GLUT4) in adipocytes, thereby enhancing glucose uptake and utilization [110] (Figure 5). Another proposed mechanism is that vitamin D binding to VDR leads to the inhibition of the TLR-4/NF-κB pathway and suppression of pro-inflammatory gene expression [111]. Moreover, higher expression of TLR-4, NF-κB, and downstream inflammatory mediators was associated with low vitamin D levels in an in vitro model of primary hepatocytes; this effect was reversed by vitamin D supplementation through the downregulation of TLR4-mediated inflammatory pathways [112]. However, the aforementioned association could be due to factors related to both NAFLD and obesity, such as dietary factors or a sedentary lifestyle [113]. Studies have shown that vitamin D supplementation leads to a decrease in CD4+ T lymphocytes and splenocytes and promotes splenocyte apoptosis by downregulating the anti-apoptotic proteins Bcl-xL and Bcl-2 [114]. In parallel, another study showed that vitamin D contributes to splenic protection and reduces the adverse effects of a high-fat diet on the spleen and immune system [115].

Beyond the role of vitamin D in glucose metabolism and insulin resistance, it is suggested that impairment of the vitamin D–VDR axis may impair gut innate immunity and contribute to liver pathology via the downregulation of Paneth cell alpha defensins, bacterial translocation, endotoxemia, low-grade inflammation, insulin resistance, and hepatic steatosis [116]. Increased gut permeability leads to bacterial translocation, which in turn induces TLRs on KCs [117]. Vitamin D suppresses the expression of TLRs in KCs and reduces inflammation [117] (Figure 5).

The binding of vitamin D to VDR results in the decreased proliferation of HSCs and contributes to the alleviation of liver fibrosis [25] (Figure 5). The interaction and co-localization of VDR and hepatocyte nuclear factor 4 α (HNF4α) in the nucleus ameliorates metabolic abnormalities [118]. Vitamin D induces intestinal fibroblast growth factor (FGF) 15 (human ortholog FGF19), which phosphorylates hepatic FGF receptor (FGFR) 4, acting on the liver and suppressing CYP7A1. Insulin and FGF15/19 inhibit forkhead transcription factor 1 (FOXO1), an important mediator of FGF and insulin signaling pathways that contributes to glucose and bile acid metabolism [119] (Figure 5).

Bozic et al. [92] reported that VDR knockout mice were protected from liver steatosis, dyslipidemia, and insulin resistance and exhibited decreased synthesis of taurine-conjugated bile acids, whereas the exposure of mice to a high-fat diet (HFD) resulted in the early induction of hepatic VDR expression in the presence of a fatty liver, followed by a long-term decrease in VDR levels in the presence of more severe inflammation and fibrosis. Moreover, VDR activation caused pro-steatotic function in hepatocytes, possibly by inducing lipogenic signaling pathways and suppressing fat oxidation [120]. Specifically, apoE^-/-^VDR^-/-^ mouse livers showed decreased gene expression of CD36, DGAT2, C/EBPα, and FGF21 and increased expression of PNPLA2, LIPIN1, and PGC1α [120] (Figure 5). In contrast, a recent study showed that vitamin D supplementation had a protective effect on HFD-induced NAFLD through VDR induction and that VDR knockout mice showed worsening of HFD- or methionine- and choline-deficiency-induced liver steatosis [118]. Moreover, the liver-specific deletion of VDR reduced the rescue effect of vitamin D on hepatic steatosis and insulin resistance in mice, demonstrating that the regulatory role of vitamin D depends mainly on VDR activation [118]. VDR activation has also been shown to increase the expression of hepatic angiopoietin-like protein 8 (ANGPTL8), a protein closely associated with NAFLD and an important regulator of triglyceride metabolism [121] (Figure 5).

To note, VDR polymorphisms have been associated with the severity of liver fibrosis in patients with biopsy-proven NAFLD [122], supporting the notion that VDR affects insulin-sensitive tissues and organs such as adipose tissue and the liver.

#### Vitamin D Supplementation and NAFLD

The relationship between vitamin D deficiency and the pathogenesis of NAFLD and the potential role of vitamin D supplementation in the treatment of NAFLD have been extensively studied [123]. The data from preclinical studies suggest that vitamin D supplementation ameliorates the histopathology of NAFLD, and vitamin D acts as a crucial anti-fibrotic agent against liver injury caused by thioacetamide [25]. In patients undergoing liver transplantation, vitamin D supplementation resulted in the prevention of acute cellular rejection [124]. However, several studies suggest that vitamin D supplementation has little effect on the pathogenesis of NAFLD related to liver injury, hepatic fat accumulation, and hepatic steatosis [125,126]. In parallel, a meta-analysis showed that vitamin D supplementation did not improve insulin resistance, liver enzymes, glucose metabolism parameters, or lipid levels in NAFLD patients [127].

The evaluation of the efficacy of 6 months of intramuscular vitamin D supplementation in improving liver-related parameters in NAFLD patients found that a single monthly intramuscular dose of cholecalciferol was effective in improving laboratory and fibroscan markers in NAFLD patients [128]. The effect of vitamin D supplementation on anthropometric and biochemical variables in NAFLD patients was investigated in a recent meta-analysis; the results showed that vitamin D significantly improved indices such as high-density lipoprotein-cholesterol, body weight, body mass index, waist circumference, serum alanine transaminase, serum fasting blood glucose, homeostatic model assessment for insulin resistance (HOMA-IR), and calcium markers, suggesting its use as an alternative treatment strategy for NAFLD patients [129].

Several studies have examined the effects of vitamin D supplementation in NAFLD patients in association with insulin resistance. A randomized double-blind clinical trial was conducted in NAFLD patients to compare the effect of calcitriol with that of cholecalciferol, using HOMA-IR as an indicator of insulin resistance. The calcitriol supplementation significantly reduced HOMA-IR compared with cholecalciferol in NAFLD patients [130]. A recent meta-analysis examined the effects of vitamin D treatment on insulin resistance in NAFLD patients, and the results were positive, suggesting that vitamin D supplementation may reduce HOMA-IR and may be appropriate as a complementary treatment for such patients [131].

A prospective randomized trial evaluated the efficacy of parenteral and oral vitamin D supplementation on serum vitamin D levels in NAFLD patients [132]. The results showed that the parenteral supplementation method was as effective as oral administration in improving serum 25(OH) vitamin D levels [132]. In addition, the defects in the enterohepatic circulation (EHC) and factors that inhibit vitamin D absorption had no significant effect on serum vitamin D levels in NAFLD patients [132]. A recent study examined the effects of supplementation with fish oil and vitamin D3 (FO + D) on serum biomarkers of NAFLD. They showed that supplementation with FO + D and FO had similar beneficial effects on markers of hepatocellular injury and plasma triacylglycerol levels in NAFLD patients [133]. In parallel, additional favorable results were observed in the group supplemented with FO + D compared with the FO group in terms of insulin levels and inflammation [133].

## 8. Vitamin D–VDR-Related Genetic Polymorphisms in Liver Disease

Genetic polymorphisms have been found to affect vitamin D levels; such variants are found in genes encoding dehydrocholesterol reductase-7 (DHCR7) [134], nicotinamide adenine dinucleotide synthetase-1 (NADSYN1) [134], the gene encoding the group-specific component (GC) for DBP [135,136], and in several genes encoding cytochrome P450 (CYP) enzymes involved in the formation of 25(OH)D and 1,25(OH)2D and in the inactivation of 1,25(OH)2D (CYP2R1, CYP27B1, and CYP24A1) [134,136].

Moreover, recent studies have reported a genetic association between VDR polymorphisms and various chronic liver diseases (AIH, PBC, HBV, and HCV infections, and HCC) [137,138,139,140,141,142,143,144,145,146]. In parallel, the progression of liver fibrosis was associated with the presence of VDR polymorphisms in PBC [138], HCV [146,147], liver cirrhosis [148], and NAFLD [138,147,149]. In particular, in PBC, the VDR polymorphisms BsmI and TaqI were associated with advanced fibrosis or liver cirrhosis [138]; in hepatitis C, the development of cirrhosis was associated with homozygosity for the dominant traits of the ApaI and FokI variants [146], the VDR bAt(CCA) haplotype was associated with the rapid progression of fibrosis [147], the GG genotype of DHCR7 rs12785878 had a significantly increased risk of liver fibrosis [150], and rs1800469 in the TGF-β1 gene was significantly associated with hepatic decompensation [83]; in liver cirrhosis, ApaI, TaqI, and BsmI polymorphisms were associated with disease severity [148]; in NAFLD, BsmI was significantly associated with the severity of liver fibrosis [122]; homozygous alleles for A1012G, BsmI, and Tru9I polymorphisms affected VDR mRNA expression, and the A1012G polymorphism reduced the antifibrotic effect of vitamin D supplementation [149]; in pediatric NAFLD, SNPs on the NADSYN1/DHCR7 (rs12785878, rs3829251) and FokI on the VDR were independently associated with increased steatosis, and the GC variant (rs4588) was associated with increased inflammation [134]; and in chronic liver disease, the rs12785878 DHCR7 polymorphism was associated with liver stiffness [151].

Beilfuss et al. [149] showed that vitamin D had no effect on the expression of TGF-β-induced profibrogenic genes in HSCs and in primary cells of NAFLD patients carrying the minor allele of A1012G SNP [149]. These results are consistent with data showing that vitamin D regulates the effects of TGF-β in VDR deficiency and is partially dependent on VDR SNPs in cultured primary human HSCs [149].

VDR polymorphisms may influence immunomodulation by affecting cytokine levels, and may play a role in liver disease progression. Considering that vitamin D inhibits Th1-mediated responses, it has been hypothesized that impaired activity of VDR-mediated signaling pathways caused by VDR polymorphisms may direct the immune response to the Th1 pathway and may contribute to the development of liver disease. However, the biological relationship between VDR polymorphisms and the alteration of vitamin D activity remains unclear. Our recent study demonstrated an association between polymorphisms of the VDR gene and cytokine levels, the severity of liver disease, and survival in patients with liver cirrhosis [148]. Specifically, we showed that the ApaI, TaqI, and BsmI polymorphisms were associated with the severity of liver cirrhosis via the immunoregulatory process. In addition, patient survival was related to the FF genotype of the FokI polymorphism, suggesting a possible protective role in liver cirrhosis [148].

## 9. Vitamin D Signaling and Gut Microbiota in Liver Disease

Vitamin D is strongly involved in the structural maintenance and functional integrity of intestinal mucosal cells and exerts antimicrobial effects [152]. In addition, vitamin D–VDR signaling has been shown to regulate immunity to gut pathogens [153,154,155,156,157]. Data from animal studies have shown that vitamin D deficiency or a defect in vitamin D-related signaling pathways impairs the gut’s innate immunity, including the downregulation of defensins expressed on Paneth cells, leading to intestinal dysbiosis, endotoxemia, and low-grade systemic inflammation, which in turn promotes the development of insulin resistance and metabolic disorders [117,158].

VDR is highly expressed in the distal region of the small intestine and in the colon; the activation of vitamins by VDR leads to the regulation of genes that control the gut physiology, homeostasis, barrier function, and inflammatory responses by cells of the secretory and immune system [159]. In patients with liver cirrhosis, vitamin D deficiency has been associated with bacterial infections [160]. A recent study examined the role of calcitriol in bacterial translocation in cirrhotic rats and showed that calcitriol administration attenuated the bacterial translocation and decreased the intestinal permeability in a thioacetamide-induced cirrhotic rat model by directly upregulating tight junction proteins in the colon (claudin-1) and small intestine (occludin) [161]. In addition, this study showed that calcitriol supplementation resulted in the enrichment of potentially beneficial taxa of the gut microbiota (*Bacteroidales*, *Muribaculaceae*, *Allobaculum, Ruminococcaceae,* and *Anaerovorax*) [161].

A recent study demonstrated the importance of vitamin D 25-hydroxylation in the small intestine in maintaining Paneth cell function [162]. Deficient hepatic 25-hydroxylation of vitamin D was demonstrated in carbon-tetrachloride-induced liver injury, resulting in the downregulation of defensins expressed in Paneth cells, gut dysbiosis, and endotoxemia [162]. Intestinal VDR knockout mice showed decreased Paneth cell defensins and lysozyme production, as well as worsening liver injury and fibrogenesis [162]. The results of this study suggest that liver injury impairs vitamin D synthesis, which in turn disrupts Paneth cell function in the small intestine and induces gut dysbiosis and liver fibrogenesis [162].

Considering that the gut microbiota is recognized as an important factor in the progression of NAFLD [163], an analysis of the gut microbiota in NAFLD rats showed that vitamin D restored the HFD-induced dysbiosis of the gut microbiota by increasing *Lactobacillus* and decreasing *Acetatifactor*, *Oscillibacter**,* and *Flavonifractor* [164]. The results of this study suggest that vitamin D may ameliorate HFD-induced NAFLD by altering the composition of the gut microbiota [164].

## 10. Conclusions

Chronic liver disease leads to liver fibrosis, cirrhosis, and eventually HCC. Although vitamin D–VDR signaling plays a key role in controlling liver homeostasis, inflammation, and fibrogenesis, it is not clear how the interplay between hepatocytes is regulated in a damaged liver. The activation of vitamin D–VDR leads to the activation of a specific signaling cascade that affects the expression of specific target genes. Understanding the molecular mechanisms underlying the regulation of liver homeostasis by vitamin D and VDR may implicate vitamin D in a broad spectrum of liver disease pathogenesis. A recent review highlighted the biological basis and mechanisms of action of vitamin D in chronic liver disease. This review presented evidence that the interplay of vitamin D and VDR regulates signaling pathways controlling the expression of antiproliferative, anti-inflammatory, and antifibrotic genes, and discussed the prominent role of vitamin D in the pathophysiology of liver dysfunction and the influence of genetic polymorphisms on inflammatory responses and fibrogenic outcomes [17]. The emerging evidence of genetic polymorphisms or other factors influencing the host response to vitamin D signaling represents another complex dimension. Considering that the currently available therapeutic strategies for liver fibrosis are limited, future studies should focus on vitamin D-mediated mechanistic and genetic insights to provide novel therapeutics for the treatment of liver disease.

## Figures and Tables

**Figure 1 ijms-23-08465-f001:**
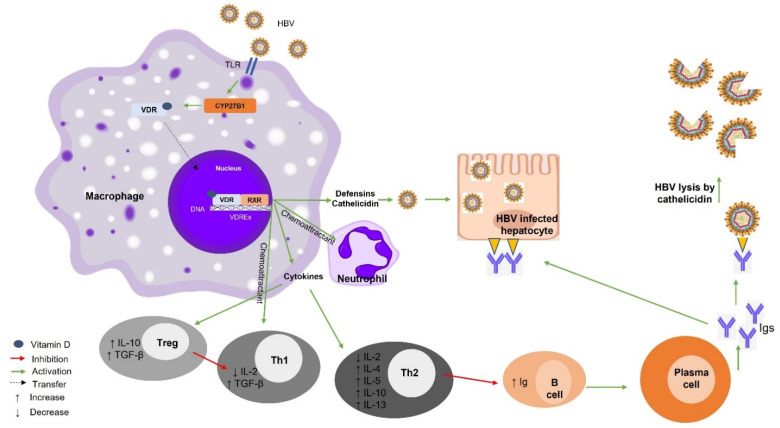
Vitamin D–VDR-mediated effects on hepatitis B infection. Binding of HBV to TLR leads to increased vitamin D production and CYP27B1 stimulation. Binding of VDR to RXR and their recruitment to VDREs leads to the secretion of defensins, cathelicidins, and cytokines. The expression of these immunomodulators leads to the transition from a pro-inflammatory response mediated mainly by Th1 cells and antibody production to an anti-inflammatory response mediated mainly by Tregs and Th2 cells. HBV, hepatitis B virus; TLR, Toll-like receptor; VDR, vitamin D receptor; RXR, retinoid X receptor; VDRE, vitamin D response element; Th1, T helper 1; Th2, T helper 2; CYP27B1, cytochrome P450, family 27, subfamily B, polypeptide 1; Tregs, T regulatory cells; Ig, immunoglobulin; IL-, interleukin; TGF-β, transforming growth factor beta.

**Figure 2 ijms-23-08465-f002:**
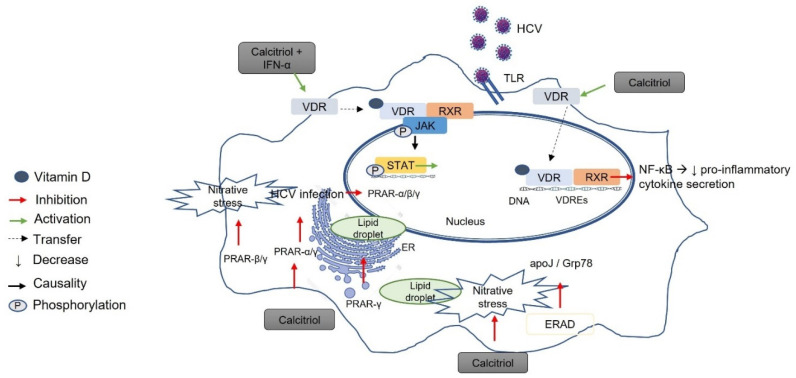
Vitamin D–VDR-mediated effects on hepatitis C infection. The binding of vitamin D to VDR modulates the HCV-induced inflammatory response by inhibiting the TLR/NF-κB pathway and simultaneously suppressing the synthesis of pro-inflammatory cytokines. The pathogenicity of HCV is modulated by calcitriol via the activation of VDR and blockade of PPAR-α/β/γ activity. The inhibitory effect of calcitriol on PPAR-α/γ leads to the suppression of viral infection, its inhibitory effect on PPAR-β/γ leads to a reduction in nitrative stress, and its inhibitory effect on PPAR-γ leads to a reduction in lipid accumulation. In parallel, the inhibition of apoJ and Grp78 via the ERAD pathway by calcitriol contributes to HCV suppression. The IFN-α-induced binding of phosphorylated STAT1 to its DNA target sequences is enhanced by calcitriol. The silencing of VDR leads to an enhanced hepatocellular response to IFN-α treatment via the JAK-STAT pathway. HCV, hepatitis C virus; TLR, Toll-like receptor; VDR, vitamin D receptor; RXR, retinoid X receptor; VDRE, vitamin D response element; PPRE, PPAR response element; ER, endoplasmic reticulum; ERAD, ER-associated degradation; apoJ, apolipoprotein J; Grp78, 78 kDa glucose-regulated protein; SVR, sustained virological response; miR155, microRNA 155; SOCS1, suppressor of cytokine signaling 1; NF-κB, nuclear factor-κB; STAT1, signal transducer and activator of transcription 1; JAK, Janus kinase.

**Figure 3 ijms-23-08465-f003:**
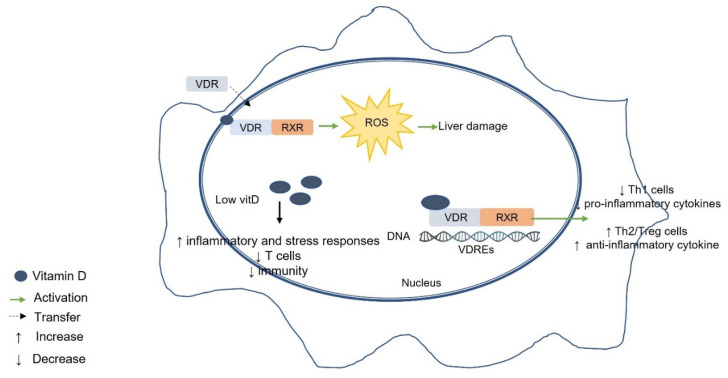
Vitamin D–VDR-mediated effects on autoimmune hepatitis (AIH). Low vitamin D levels and increased inflammatory and stress responses lead to decreased T-cell subpopulations and impaired immunity in AIH patients. VDR signaling leads to a rapid increase in ROS, which could be detrimental to the liver in AIH. VDR, vitamin D receptor; RXR, retinoid X receptor; VDRE, vitamin D response element; Th1, T helper 1; Th2, T helper 2; Tregs, T regulatory cells; ROS, reactive oxygen species.

**Figure 4 ijms-23-08465-f004:**
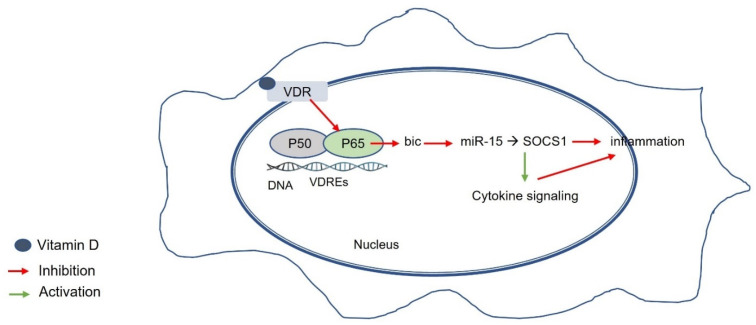
Vitamin D–VDR-mediated effects on primary biliary cholangitis (PBC). Vitamin D–VDR signaling alleviates inflammation by targeting the miRNA155-SOCS1 axis in PBC. VDR/miRNA155-modulated SOCS1 expression is reduced in PBC, resulting in impaired regulation of cytokine signaling. VDR, vitamin D receptor; VDRE, vitamin D response element; miR155, microRNA 155; SOCS1, suppressor of cytokine signaling 1.

**Figure 5 ijms-23-08465-f005:**
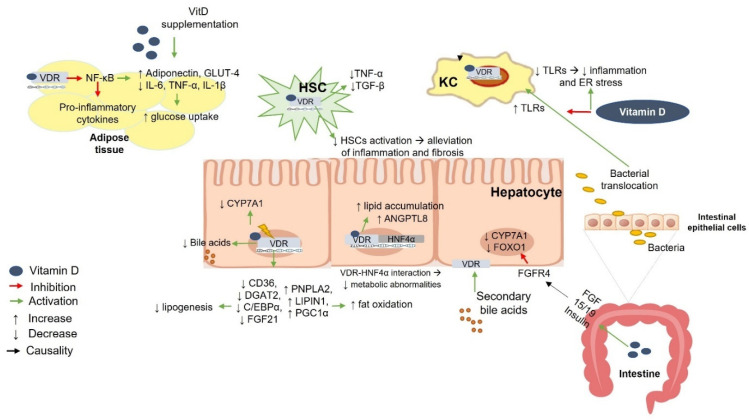
Vitamin D–VDR-mediated effects on nonalcoholic fatty liver disease (NAFLD). Vitamin D acts on adipose tissue by inhibiting NF-κB transcription and suppressing the expression of inflammatory cytokines. Higher adiponectin secretion improves insulin resistance by promoting transcriptional regulation and translocation of GLUT4 into adipocytes, leading to improved glucose uptake. Increased gut permeability leads to translocation of bacteria, which in turn induce TLRs on KCs. Vitamin D suppresses the expression of TLRs in KCs and reduces inflammation. Binding of vitamin D to VDR leads to reduced proliferation of HSCs, thereby contributing to the alleviation of liver fibrosis. Co-localization of VDR with HNF4α in the nucleus ameliorates metabolic abnormalities. Vitamin D upregulates intestinal FGF15 (human ortholog FGF19), which phosphorylates hepatic FGFR4 in the liver to inhibit CYP7A1. Insulin and FGF15/19 suppress FOXO1, a key mediator of the FGF and insulin pathways, contributing to bile acid and glucose metabolism. Stimulation of VDR accelerates lipid accumulation in the liver and increases the expression of ANGPTL8, a key modulator of triglyceride metabolism. The deletion of VDR and apolipoprotein E results in decreased gene expression of CD36, DGAT2, C/EBPα, and FGF21 and increased expression of PNPLA2, LIPIN1, and PGC1α in mouse livers. VDR, vitamin D receptor; TLR, toll-like receptor; ER, endoplasmic reticulum; GLUT4, glucose transporter 4; FGF15/19, fibroblast growth factor 15/human ortholog 19; FGFR4, FGF receptor 4; FOXO1, forkhead transcription factor 1; CYP7A1, cholesterol 7a-hydroxylase; HNF4α, hepatocyte nuclear factor 4 α; ANGPTL8, angiopoietin like 8; CYP27B1, cytochrome P450, family 27, subfamily B, polypeptide 1; HSC, hepatic stellate cells; KC, Kupfer cell; NF-κB, nuclear factor-κB; IL-, interleukin; TNF-α, tumor necrosis factor α; TGF-β, transforming growth factor beta; CD36 (cluster of differentiation 36); DGAT2, diacylglycerol O-acyltransferase 2; C/EBPα, CCAAT/enhancer binding protein α; FGF21, fibroblast growth factor 21; PNPLA2, patatin-like phospholipase domain-containing protein 2; PGC1α, peroxisome-proliferator-activated receptor gamma coactivator 1-alpha.

## Data Availability

Not applicable.

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
