# Peer review of "Vitamin D–VDR Novel Anti-Inflammatory Molecules—New Insights into Their Effects on Liver Diseases"

_ijms, 2022, doi:10.3390/ijms23158465_

Round 1

Reviewer 1 Report

The manuscript in question reviews the complex interaction between immune and liver cells in the maintenance of liver homeostasis and the development of liver injury. The advances in the molecular mechanisms of vitamin D-VDR signaling are also critically discussed in terms of liver disease development and progression. The most recent acquisitions on the role of vitamin D and VDR-associated genetic variants in regulating the progression and severity of liver disease are also addressed. The review is a timely one and it is well written and easy and enjoyable to read. I don’t have any particular point to raise, I just suggest to use the same graphic symbol in indicating Vitamin D in the cartoons composing the figures.

Author Response

Reviewer 1

The manuscript in question reviews the complex interaction between immune and liver cells in the maintenance of liver homeostasis and the development of liver injury. The advances in the molecular mechanisms of vitamin D-VDR signaling are also critically discussed in terms of liver disease development and progression. The most recent acquisitions on the role of vitamin D and VDR-associated genetic variants in regulating the progression and severity of liver disease are also addressed. The review is a timely one and it is well written and easy and enjoyable to read.

Comment 1: I don’t have any particular point to raise, I just suggest to use the same graphic symbol in indicating Vitamin D in the cartoons composing the figures.

Response to comment 1:

We have revised the figures to use the same graphic symbols throughout.

Reviewer 2 Report

The Authors pointed out several liver diseases, including chronic HBV, chronic HVC, autoimmune hepatitis, primary biliary cholangitis and non-alcoholic fatty liver disease, associated with vitamin D. 

There is no information about the selection criteria of the above mentioned diseases.

The presented data are not novel. 

What's more, the are published papers with a more comprehensive overview on vitamin D and liver disease.

Author Response

Reviewer 2

The Authors pointed out several liver diseases, including chronic HBV, chronic HVC, autoimmune hepatitis, primary biliary cholangitis and non-alcoholic fatty liver disease, associated with vitamin D. 

Comment 1:

There is no information about the selection criteria of the above-mentioned diseases.

Response to comment 1:

We have now introduced the selection criteria for the studies included in this work (cf. Materials and Methods, p.2, lines 58-77)

Comment 2: The presented data are not novel. What's more, the are published papers with a more comprehensive overview on vitamin D and liver disease.

Response to comment 2:

We are aware that this particular topic has been discussed before, but the great importance of the vitamin D-VDR effect in the context of liver disease deserved much more attention. For this reason, we decided to highlight this topic in this review. The association of the vitamin D-VDR complex with liver dysfunction shows great potential for clinical application and supports its use as a prognostic index and diagnostic tool. In parallel, although there are many data on the association between vitamin D deficiency in patients with liver disease and liver dysfunction, disease severity, and poor prognosis, there is little understanding of the mechanisms by which vitamin D and its receptor (VDR) contribute to the regulation of innate and adaptive immune responses. A complete understanding of the molecular mechanisms involved in vitamin D-VDR signaling and the functional significance of VDR signaling in specific cell populations in liver disease will help to better understand the enormous amount of data that is still lacking. Furthermore, in our review, we have attempted to comprehensively present and discuss the current literature, with explanatory figures summarizing the associated mechanisms in each liver disease. Finally, the addition of the interplay between vitamin D and the gut microbiota in liver disease and the therapeutic value of vitamin D supplementation, as suggested by Reviewer 3, has added pertinent information to this review. opening new future research directions in this field.

Reviewer 3 Report

The review proposed by Ioanna Aggeletopoulou et al. concerns a topic of great interest and actuality. The Authors investigated the immunomodulatory and anti-inflammatory effects of Vitamin D, providing interesting pathophysiological information in the context of specific liver diseases, such as hepatitis B and C, autoimmune hepatitis, primary biliary cholangitis and nonalcoholic fatty liver disease (NAFLD).

The text is full of detailed information, the paragraphs systematically address the main theme, and the figures are very clear and explanatory. Overall, the manuscript is well written. I suggest only some changes and implementations:

1)      Introduction:  a more detailed insight into the role of vitamin D and liver-spleen axis is suggested (DOI: 10.1080/10408398.2017.1353479)

2)      As part of the emerging mechanisms, the microbiota is assuming a growing role in the pathogenesis of some chronic diseases, especially metabolic diseases, due to production of specific metabolites effective in the modulation of the immune system activity. Although there is few evidence on the impact of the microbiota on the immunomodulation and liver diseases, it could be interesting to address this issue (DOI: 10.1016/j.abb.2021.108894; doi: 10.1002/mnfr.202000937. doi: 10.1152/ajpgi.00021.2020. doi: 10.1152/ajpgi.00286.2019. doi: 10.1080/10408398.2020.1792826.)

3)      The physiopathological role of vitamin D in liver diseases is well emphasized in the text. However, to make the description more complete and to also to offer information on possible therapeutic implications, I would suggest enriching each paragraph with studies relating to vitamin D supplementation. I also recommend underlining the possible immunomodulatory role of vitamin D supplementation in some liver diseases.

4)      A minor linguistic revision of the text is recommended

Author Response

Reviewer 3

The review proposed by Ioanna Aggeletopoulou et al. concerns a topic of great interest and actuality. The Authors investigated the immunomodulatory and anti-inflammatory effects of Vitamin D, providing interesting pathophysiological information in the context of specific liver diseases, such as hepatitis B and C, autoimmune hepatitis, primary biliary cholangitis and non-alcoholic fatty liver disease (NAFLD).

The text is full of detailed information, the paragraphs systematically address the main theme, and the figures are very clear and explanatory. Overall, the manuscript is well written. I suggest only some changes and implementations.

Comment 1: Introduction: a more detailed insight into the role of vitamin D and liver-spleen axis is suggested (DOI: 10.1080/10408398.2017.1353479)

Response to comment 1:

We have provided more detail on the role of vitamin D and the liver-spleen axis in liver disease as proposed. We have added information explaining the association between the liver-spleen axis and inflammation in the Introduction (cf. p.1, lines 33-40), and information describing studies supporting the association between vitamin D and the liver-spleen axis in the Vitamin D-VDR and NAFLD section (cf. p.11-12, lines 454-466).

Comment 2: As part of the emerging mechanisms, the microbiota is assuming a growing role in the pathogenesis of some chronic diseases, especially metabolic diseases, due to production of specific metabolites effective in the modulation of the immune system activity. Although there is few evidence on the impact of the microbiota on the immunomodulation and liver diseases, it could be interesting to address this issue (DOI: 10.1016/j.abb.2021.108894; doi: 10.1002/mnfr.202000937. doi: 10.1152/ajpgi.00021.2020. doi: 10.1152/ajpgi.00286.2019. doi: 10.1080/10408398.2020.1792826.)

Response to comment 2:

The enhancement of the innate immune system by acting as an immune modulator, the anti-inflammatory effects, and the important contribution of vitamin D to maintaining the integrity of the gut barrier and regulating the gut microbiota provide plausible mechanisms for how vitamin D may influence the pathogenesis and progression of immune-related diseases. This topic has attracted considerable interest, particularly in recent years. As reported by the reviewer, the microbiota appears to be closely linked to the development of some chronic diseases, particularly metabolic diseases. Therefore, we have added a section titled “Vitamin D signaling and gut microbiota in liver disease” describing the latest data on this topic (cf. p.15-16, lines 628-667).

Comment 3: The physio pathological role of vitamin D in liver diseases is well emphasized in the text. However, to make the description more complete and to also to offer information on possible therapeutic implications, I would suggest enriching each paragraph with studies relating to vitamin D supplementation. I also recommend underlining the possible immunomodulatory role of vitamin D supplementation in some liver diseases.

Response to comment 3:

The aim of the current study focused on the molecular and cellular mechanisms underlying the pathogenesis of various liver diseases. In parallel, we discussed the interplay of liver and immune cells in the maintenance of liver homeostasis and the development of liver injury, the role of vitamin D - VDR in the development and progression of liver disease, and the influence of vitamin D-VDR-associated genetic variants in modulating the occurrence and severity of liver disease. However, enrichment of this manuscript with studies on vitamin D supplementation and the possible immunomodulatory role of vitamin D supplementation in some liver diseases would be a very interesting addition. Therefore, in the revised manuscript, we have added information on vitamin D supplementation in each liver disease studied, as suggested by the reviewer (cf. p.6, lines 236-248; p.8-9, lines 328-383; p.10, lines 402-415; p.11, lines 434-442; p.13-14, lines 530-583).

To the extent that the reviewer refers to the possible immunomodulatory role of vitamin D supplementation in liver disease, this has already been mentioned in the present review. Where appropriate, data from experimental in vitro studies or animal studies have been discussed. For HCV, there are reports on p.6, lines 259-265, p.7-8, lines 285-310, and for NAFLD on p.11, lines 449-457, 460-466, 516-519.

Comment 4:

A minor linguistic revision of the text is recommended

Response to comment 4:

This manuscript has been edited for correct English language, grammar, punctuation, spelling, and general style by one of the authors, who is a native English speaker with extensive experience in writing scientific manuscripts.

Round 2

Reviewer 2 Report

The Authors provided a revised version of the manuscript.

Selection criteria and search strategies were clearly indicated.

I agree with the Authors that one of the superlatives of this paper are colourful Figures indicating mechanism of interference between liver (disease) and vitamin D.

In its actual form I found the article worth publishing.